# Production of Transgenic Silkworm Using Anti-Serum Against Diapause Hormone in Diapause Strains of Silkworm, *Bombyx mori*

**DOI:** 10.3390/ijms26157604

**Published:** 2025-08-06

**Authors:** Keiro Uchino, Megumi Sumitani, Tetsuya Iizuka, Hideki Sezutsu

**Affiliations:** Institute of Agrobiological Sciences, National Agriculture and Food Research Organization, 1-2 Owashi, Tsukuba 305-8634, Japan; kasashima.megumi510@naro.go.jp (M.S.); iizuka.tetsuya074@naro.go.jp (T.I.); sezutsu.hideki246@naro.go.jp (H.S.)

**Keywords:** non-diapause strain, microinjection, anti-serum against diapause hormone, *Bombyx mori*

## Abstract

In general, the silkworm, *Bombyx mori*, has a diapause trait in its eggs. Therefore, transgenic silkworm can be produced by embryonic microinjection using eggs laid by a non-diapause strain in *B. mori.* In this study, we performed microinjection using eggs of diapause strains which have good characteristics for industrial use, such as a big cocoon, thin and smooth silk, and tolerance against disease due to the growing industrial use of transgenic silkworms. For the conversion of egg diapause traits from diapause to non-diapause types, we used anti-serum against the diapause hormone of *B. mori* (BmDH), which was injected into maternal pupae, producing non-diapause eggs at a high rate. Finally, we attempted microinjection using three diapause strains with different voltinism (i.e., number of generations of an organism in a year) and were able to successfully produce transgenic silkworms in all three of them, demonstrating that our method is applicable to a wide range of silkworm strains with a diapause trait.

## 1. Introduction

The silkworm, *Bombyx mori*, is a beneficial insect that has recently been used not only for academic studies [1,2,3,4,5,6,7,8] but also for practical purposes, such as the production of useful recombinant proteins in industrial, biopharmaceutical, and biomaterial fields through transgenic silkworm [9,10,11,12,13,14,15,16,17,18,19,20,21,22]. In general, the silkworm has a diapause trait in the egg. Therefore, transgenic silkworms have been produced by embryonic microinjection using a non-diapause strain that has a mutant gene, *pnd*, one of the genes responsible for the diapause trait [23]. However, other strains have some beneficial traits for industrial use, such as big cocoons in abundant production of recombinant protein, but they usually have a diapause trait. The diapause trait is induced by a diapause hormone synthesized in the suboesophageal ganglion (SG) [24], which is a neuropeptide of 24 amino acids with amidation at the C-terminus: TDMKDESDRGAHSERGALWFGPRL-amide (NH_2_) (diapause hormone precursor, GenBank: BAA03755.1) in *B. mori* [25]. However, there are two types, BomDH-I [19-Trp] and BomDH-I [19-Cys], where the 19th position was tryptophan and cysteine, respectively [26]. The hormone synthesized in the SG is released via the corpus cardiacum–corpus allatum complex into the blood [27] and transduces the signal to the oocyte via the diapause hormone receptor that induces embryonic diapause [28], thereby inducing embryonic diapause in the progeny. Embryonic diapause induction can be suppressed by immersion in HCl solution at around 20 h post-oviposition, but the mechanism of the induction is not known. Zhao et al. attempted microinjections using eggs treated by the acid. Nevertheless, the hatching rate after the microinjections was very low, which showed that they resulted in heavy damage to the eggs [29]. Another method involves incubating maternal embryos under constant darkness at 15 °C during embryonic development (15 °C-IME-DD). This treatment is known to suppress the secretion of the diapause hormone (DH) in the maternal generation, resulting in the production of non-diapause eggs in the subsequent generation [30]. This method is effective for inducing the production of non-diapause eggs in bivoltine or polyvoltine strains, but not in univoltine strains. In fact, many silkworm strains are univoltine, meaning that they exhibit a strong diapause trait. There are several other ways to produce non-diapause eggs: dimethyl sulfoxide (DMSO) [31], oxygen gas [32], corona discharge [33], the ablation of SG, and the transplantation of the ovary to a male [34]. However, each of these methods presents certain limitations in terms of efficiency, reproducibility, and technical complexity. Therefore, we aimed to establish a more practical and reliable approach. Dr. Shiomi reported that non-diapause eggs could be produced by injecting anti-serum against *B. mori* diapause hormone (BmDH) into female pupae in diapause strains but not by nonimmune serum [35]; therefore, we utilized this approach. We utilized non-diapause eggs that were obtained by injecting anti-serum against *B. mori* diapause hormone (BmDH) into female pupae to produce transgenic silkworm in diapause strains. As a result of this anti-diapause hormone serum treatment (aDHST), the injected females laid non-diapause eggs at a high rate, not only in bivoltine strains but also in univoltine strains. Using these non-diapause eggs, we then performed embryonic microinjection and successfully produced transgenic silkworms. Our method using anti-diapause hormone serum is highly efficient and does not require any special equipment excluding conventional microinjection devices, which makes it a practical approach.

## 2. Results

### 2.1. Sensitivity of Different Diapause Strains to 15 °C-IME-DD on the Non-Diapause Trait

The diapause of some bivoltine strains in *B. mori* can suppress the diapause in the progeny by keeping the eggs at 15 °C-IME-DD in the parental generation, but this does not occur in many univoltine strains [30]. Therefore, we examined the non-diapause sensitivity of eggs by 15 °C-IME-DD in Kosetsu and C515, which have different diapause characteristics; Kosetsu is bivoltine and C515 is univoltine. A comparative experiment was performed between eggs treated in 15 °C-IME-DD, and in 25 °C. Our study showed that Kosetsu was sensitive to 15 °C-IME-DD which raised non-diapause eggs with 69.2% hatchability, while C515 was not sensitive to the same condition: 0% hatchability (Table 1). However, both strains had no or little sensitivity to the temperature of 25 °C and then caused diapause; the hatchability at this temperature was 0.1% in Kosetsu and 0% in C515 (Table 1).

### 2.2. Injection of Anti-BmDH Serum into Pupa and Production of Non-Diapause Eggs

To investigate the effects of anti-BmDH serum for the production of non-diapause eggs, we injected 7 μL serum in each female pupa of Kosetsu and C515 at the 1st day after ecdysis. The polyclonal anti-BmDH serum, serum [19-Cys], was kindly offered by Dr. Shiomi, Shinshu University, which was produced by immunizing rabbits with the synthetic diapause hormone (BomDH-I [19-Cys]) of the silkworm, *B. mori* [35]. The injected pupae were kept at 25 °C until eclosion, the emerged female moths were crossed with male moths uninjected with the anti-serum, and the eggs laid were examined for the diapause trait (first half of Figure 1).

The individuals after aDHST could normally develop into adults just like the controls, which were non-injected individuals, and lay eggs to the same degree as the untreated groups, and their eggs’ color was changed to white which means that their diapause characteristic turned into non-diapause. However, the controls developed normal dark brown pigmentation, characteristic of diapause, which was derived from ommochrome pigments [36] (Figure 2).

We investigated the hatchability of progeny eggs derived from female pupae after aDHST. Eggs collected in the first hour of laying in the serum-injected moths were used for embryonic microinjection, and the other eggs in the second collection were used for the examination of hatchability. As a result, the hatchability rates of eggs derived from the serum-injected moths were 89.9% in Kosetsu and 70.5% in C515, while those in the untreated moths were 0.05% in Kosetsu and 0% in C515 (Table 2).

### 2.3. Production of Transgenic Silkworms

To investigate the influence on the production of transgenic silkworm using non-diapause eggs made from the injection of anti-BmDH serum into pupae, we chose three host strains for embryonic microinjection: the Chinese strains Kosetsu and C515 and the Japanese strain MN2. Kosetsu and C515 have normal silkworm characteristics, such as colored eggs derived from ommochrome pigments [36] and black skin in the first instar larvae derived from melanin pigment, whereas MN2 has white eggs and a distinctive light brown skin in the first instar larvae because of the white egg 1 mutant. In the injection of the serum, we used two anti-BmDH sera; one was serum [19-Cys] and the other was a newly produced one, serum [19-Trp], which was produced by immunizing rabbits with a chemically synthesized antigen, TDMKDESDRGAHSERGALWFGPRL-NH_2_, corresponding to BomDH-I [19-Trp] (see the details in “Materials and Methods”). Those strains and sera were used as follows: 7 μL of serum [19-Cys] per pupa was injected in Kosetsu and C515, and 90 μL of serum [19-Trp] was injected in MN2. Despite the large difference in the amount injected, there was no effect on the survival rate of the pupae. Additionally, the hatchability of eggs derived from female moths injected with the sera was equal to that of eggs produced in the conventional way, which uses the non-diapause strain pnd-w1 [23].

Different vectors were used in the embryonic microinjection of each strain to produce transgenic silkworms. The piggyBac vector pBac(3xP3DsRed2) [37] was microinjected at a final concentration of 0.2 μg/μL in the Kosetsu strain, which can express the DsRed2 driven by 3xP3 promoter exhibiting red fluorescence in the silkworm eye. pBac(3xP3DsRed2) and pBac[IE1Nat3xP3GFP] were used in a mixture at a final concentration of 0.1 μg/μL each in the C515 strain. pBac[IE1Nat3xP3GFP] has two visible marker genes, the *IE1Nat* gene and the *3xP3GFP* gene. The former ubiquitously expresses arylalkylamine-N-acetyl transferase via the baculovirus immediate early 1 gene promoter, which can change the color of newly hatched first-instar larvae from black to light brown [38]. The latter can express enhanced green fluorescent protein (GFP) driven by 3xP3 promoter exhibiting green fluorescence in the silkworm eye. pBac3xP3GFPafm [39] was used at a final concentration of 0.2 μg/μL in the MN2 strain, which can exhibit green fluorescence in the silkworm eye. The hatchability after embryonic microinjection was 76.3% in Kosetsu, 41.1% in C515, and 62.4% in MN2 (Table 3).

As a result of screening transgenic silkworms, we found many individuals expressing the fluorescent gene in each strain (Figure 3). The acquisition rate of transgenic silkworms was 10.2% in Kosetsu, 5.7% in C515, and 71.4% in MN2 (Table 3).

### 2.4. Genomic Southern Blotting Analysis of Transgenic Lines

We performed Southern blotting analysis using the probes shown in Figure 4A to examine that the insertion events in each transgenic silkworm expressing marker genes happened independently. Consequently, different signals were detected in each transgenic individual (Figure 4B).

## 3. Discussion

We researched the improvement in embryonic microinjection using diapause strains in *B. mori*. Microinjection is usually performed using non-diapause and polyvoltine strains, such as pnd-w1 [23,40] and Nistari [41], because the diapause trait prevents the smooth production of transgenic silkworms. However, many strains of the silkworm, with various characteristics including suitability for practical use, exhibit the diapause trait in their eggs. Researchers have devised methods to prevent the embryonic diapause trait of diapause strains using treatments such as hydrochloric acid (HCl) [31], DMSO [31], oxygen gas [32], larval ovary transplantation into males [34], low-temperature treatment with constant darkness [29], and anti-BmDH serum injection into maternal female pupae [35]. The HCl treatment (HT) is the most popular way to suppress the diapause, where one day after egg laying, the eggs are usually sunk into diluted HCl solution (specific gravity at 15 °C: 1.1100). However, there is a gap because microinjection uses eggs within eight h after egg laying [42]. Zhao et al. performed embryonic microinjection in bivoltine diapause strains using the HCl treatment within 3 h of oviposition (W3HO-HT) and succeeded in producing transgenic silkworms; nevertheless, both hatchability and transgenesis were very low, with 3.4–4.6% in W3HO-HT and 3–4 positive broods, respectively [29]. Moreover, they also researched the low-temperature treatment, 15 °C-IME-DD, and found that the hatchability and transgenesis rate were 9.1–47.5% and 1–3 broods, except for one result in the seven experiments, which had 10 broods [29].

With the current efficiency of transformation in the silkworm, we have considered it important to obtain a certain hatchability rate to steadily obtain transgenic silkworms. Therefore, we decided to look for another method to suppress the diapause with treatment in the maternal generation and high hatchability but excluding 15 °C-IME-DD because it might not be useful for univoltine strains [30]. Therefore, we focused on anti-diapause hormone serum treatment (aDHST), which showed high hatchability and transgenesis rate: 41.1–76.3% and 2–15 broods (5.7–10.2% of total broods). The 15 °C-IME-DD method may be effective in the case of bivoltine strains; in fact, all six strains used in Zhao et al.’s experiments were bivoltine. Notably, our method of aDHST produced non-diapause eggs at a high rate even in the univoltine C515 strain, which cannot easily be changed into non-diapause by the 15 °C-IME-DD method, and then succeeded in producing transgenic silkworms (Figure 2 and Figure 3; Table 2 and Table 3); the effect of aDHST was also shown in all the other strains, including univoltine strains such as C514 and J604 (Table A1).

In this study, we used two anti-BmDH sera: serum [19-Cys] and serum [19-Trp]. Although they worked as neutralizing antibodies to BmDH, there was a significant difference in effect between them; the effect of the serum we produced, serum [19-Trp], was less than that of the serum [19-Cys] from Dr. Shiomi (as seen in the comparison between the data for Kosetsu in Table 2 and Table A1). However, the titers of both polyclonal antibodies were similar, as shown in Figure A2. It may be responsible for the characteristics of polyclonal antibodies. As shown in Table A1, the sensitivity to serum [19-Trp] varied between the strains; however, the appearance rate of non-diapause eggs by the injection of the anti-BmDH serum tended to increase depending on the dose (Table A1), and thus we could compensate for the lack of the effect. Additionally, the injection sites were effective not only in the thorax but also in the abdomen of the pupae (near the ovaries).

Kosegawa et al. researched the sensitivity to low temperature in the induction of non-diapause eggs in Japanese, Chinese, and European local silkworm strains in detail, using 48 (30/18), 46 (30/14), and 27 (20/0) strains, respectively; the numbers in parentheses indicate numbers of uni- versus bivoltine strains [30]. As a result, most Japanese and Chinese bivoltine strains produced non-diapause eggs by the 15 °C-IME-DD treatment at a high rate of 84.2–100%. However, 1 among 34 bivoltine strains did not produce non-diapause eggs. The embryonic diapause of bivoltine strains is sensitive not only to temperature but also to photoperiod and diets in the maternal generation [43], which is also proved by that Kosetsu’s hatchability at 25 °C was not actually 0% but rather 0.05–0.1% (Table 1 and Table 2). However, this is almost never the case in univoltine strains; nearly half of the univoltine strains produced only diapause eggs: 12 (40%) in Japanese, 14 (46.7%) in Chinese, and 16 (43.2%) in European strains by the 15 °C-IME-DD treatment [30]. These data show that there are many strains that could not produce non-diapause eggs by 15 °C-IME-DD, especially in univoltine strains. Thus, the importance of our results is evident.

In addition, we have developed different embryonic microinjection methods so far, such as DMSO [42], corona discharge [43], and the ablation of SG [44], using diapause strains. However, these methods also require extra treatments just before microinjection, as well as special equipment, and result in poor egg production, among other limitations. Of course, there are some problems with our method using aDHST too. As there is a difference between serum [19-Cys] and serum [19-Trp], the titer of antibodies could vary from lot to lot in the production of serum. Secondly, the cost of outsourcing antibodies is relatively high. Dr. Shiomi said that even antibodies made with the same antigen that had been frozen and stored showed lower activity than those used in this study; the serum given to us exhibited an especially high activity. Additionally, there were no differences in activity between anti-sera derived from BomDH-I [19-Trp] and BomDH-I [19-Cys] (personal communication). The advantages of our method in this study are that it is highly efficient and does not require any special equipment or additional work during microinjection. Taking together with the other methods described above, our method makes a valuable contribution that could broaden into diverse research applications for *Bombyx mori*.

## 4. Materials and Methods

### 4.1. Strain and Rearing

We used three main diapause strains in this study—Kosetsu, C515, and MN2—with various characteristics. Kosetsu and MN2 are bivoltine; C515 is univoltine. Kosetsu and C515 are Chinese strains; and MN2 is a Japanese strain. Additionally, MN2 has a white eye and egg color, derived from the *w1* gene mutant which was bred from J137. Kosetsu is reported in ref. [30], and MN2 and J137 are reported in ref. [45]. Those larvae were reared on a commercial diet including mulberry leaf powder SilkMate PS (Nosan Corporation, Kanagawa, Japan), with a photoperiod of 12 h in light and 12 h in darkness, at 28 °C for the 1st to the 4th and at 25 °C for the 5th instar larvae without control of humidity (around 40 to 60%). In addition to the above three strains, we used eight strains as sub-strains, as shown in Table A1: J137, J603, J604, C510, C514, DH6 [46], p50T [47], and w1 (with normal *pnd* gene). J137, J603, and J604 are Japanese strains; C510 and C514 are Chinese strains. All the above strains were maintained in the National Agriculture and Food Research Organization (NARO), Japan.

### 4.2. 15 °C-IME-DD Treatment

The 15 °C-IME-DD treatment was performed as follows. Eggs laid on the same paper were separated into two groups; one was put in a metallic box to protect them from light, and then both were incubated in the incubator set at 15 °C. The development of embryos was estimated by observing the development of the unenclosed eggs. The eggs in the metallic box were moved out of the box after black head capsules were formed in the developing eggs. After hatching, the silkworms were reared in the normal conditions described in Section 4.1 and then the adult moths were crossed in a sib-mating manner and eggs were collected to examine the change in the diapause trait.

### 4.3. Preparation of Plasmid DNA

The helper plasmid pHA3PIG [23] and the vector plasmids pBac(3xP3DsRed2) [37], pBacIE1-NAT/3xP3EGFP [38], and pBac{3xP3-EGFPafm} [39] were used in this study. These plasmid DNAs were extracted using a HiSpeed Plasmid Midi Kit (QIAGEN, Tokyo, Japan) and dissolved in injection buffer, 0.5 mM phosphate buffer (pH 7.0), and 5 mM KCl [23].

### 4.4. Preparation of Non-Diapause Eggs and Embryonic Microinjection

We procured anti-serum including polyclonal antibody against the diapause hormone of *B. mori*, which was produced through a commercial company (MEDICAL & BIOLOGICAL LABORATORIES Co., Ltd., Tokyo, Japan) as follows. The antigen of the polypeptide (BomDH-I [19-Trp]) with cysteine at the N-terminus and amide at the C-terminus CTDMKDESDRGAHSERGALWFGPRL-NH was chemically synthesized, purified, and conjugated with bovine serum albumin (BSA) according to Shiomi’s protocol [35]. The anti-BmDH serum was produced by immunizing rabbits with the above antigen. Non-diapause eggs were produced by the injection of the serum using a 25- or 29-gauge needle (SS-01T2525 or SS-10M2913 TERUMO, Tokyo, Japan) as follows: the anti-BmDH serum was injected in the amount of 7 to 90 μL into each maternal pupa of diapause strains within 24 h after ecdysis to produce non-diapause eggs in the next generation. The embryonic microinjection was performed according to refs. [23,40], excluding the cooling treatment of eggs [40]. The female moths subjected to aDHST were crossed with male moths without aDHST for three hours and the couples were kept at 5 °C for 1–3 days until microinjection. The eggs were collected as shown in Figure 1. The female moths were put on paper with starch to lay eggs. After laying, the paper was put in water to collect the eggs and then the eggs were aligned on the glass plate for microinjection. The injected pupae were kept at 25 °C until eclosion, the emerged female moths were crossed with male moths uninjected with the anti-serum, and the eggs laid were used for the examination of the diapause trait and embryonic microinjection (first half of Figure 1).

### 4.5. Screening of Transgenic Silkworms

The moths grown to adults after microinjection (G_0_) were single-mated among them (sibling mating) or the remaining moths were mated with each host strain, and then the progeny eggs (G_1_) were collected from each female moth (brood). The collected eggs were immersed in the following acid solution for 90 min: HCl diluted with distilled water, with a specific gravity of 1.1100 at 15 °C. A fluorescence binocular microscope (SZX16, Olympus, Tokyo, Japan) with a DsRed and GFP filter (RFP1: SZX2-FRFP1, GFP: SZX2-FGFP, Olympus, Tokyo, Japan) was used to screen transgenic animals in the eggs.

### 4.6. Southern Blotting Analysis

The genomic DNA samples were extracted from moths in G_1_ by SDS-phenol extraction [48] and each 2 μg of them was digested by *Msp* I (100 units/μL, NEB, Tokyo, Japan) or *Bgl* II (10 units/μL, Takara, Tokyo, Japan) for Southern blotting analysis. The probe for the genomic DNA of transgenic silkworms derived from pBac(3xP3DsRed2) was amplified as a template of the right arm region of *piggyBac* inverted terminal repeat (ITR) by PCR [49]. The probe (726bp) for the genomic DNA of transgenic silkworms derived from pBac{3xP3-EGFPafm} was amplified as a template of the EGFP coding region by PCR with the following primer sets: KS113, ATGGTGAGCAAGGGCGAGGAGCTGT, KS002, GCTAGCTTACTTGTACAGCTCGTCCA. Then, those primers were labeled using the AlkPhos Direct Labelling and Detection System with CDP-Star (GE Healthcare Co., Tokyo, Japan). The detection of targets was performed using the chemiluminescence imaging system LAS-3000 (Fujifilm Co., Tokyo, Japan) and FUSION FX7 (Vilber Lourmat, Marne-la-Vallée Cedex, France).

### 4.7. Dot Blotting Analysis

The BomDH-I [19-Trp] synthesized by Japan Bioserum Co., Ltd. (Hiroshima, Japan) was diluted with TBS-T (50 mM Tris-Cl, pH 7.6, 150 mM NaCl, 0.1% (*v*/*v*) Tween-20), spotted each 4 μL onto the PVDF transfer membrane (Amersham Hybond-P, RPN2020F: A, GE Healthcare Co., Tokyo, Japan), which was immersed in methanol for a minute and washed with water for five minutes in advance. Tris Buffered Saline (TBS) Tablets (Takara, Tokyo, Japan) were used to make TBS buffer: two tablets/L of water. The dried membrane was treated with TBS-T containing 5g (*w*/*v*%) of skim milk (Wako, Osaka, Japan) for blocking non-specific sites for an hour. Three types of rabbit serum were used to treat the membrane for each an hour; serum [19-Trp] and serum [19-Cys] was used as primary antibodies at a dilution ratio of 1/1000: nonimmunized, and the horseradish peroxidase linked anti-rabbit IgG (#NA934, GE Healthcare Co., Tokyo, Japan) was used as a secondary antibody at a dilution ratio of 1/50,000. The membrane was washed three times with TBS-T for each ten minutes between each treatment. Finally, the signal was detected using the ECL Prime Western Blotting Detection Reagent (Cytiva RPN2236, Amersham, GE Healthcare Co., Tokyo, Japan) and FUSION FX7 (Vilber Lourmat, Marne-la-Vallée Cedex, France).

## 5. Conclusions

In this study, we present a method to change diapause eggs into non-diapause eggs using anti-serum against the diapause hormone, successfully producing transgenic silkworms with high hatchability and transgenesis rates. Our results show that the proposed method successfully produces transgenic silkworms even in univoltine strains.

## 6. Patents

We have obtained a patent based on our research results in Japan: Patent No. 6765803. The patent holders are all included in this paper as co-authors.

## Figures and Tables

**Figure 1 ijms-26-07604-f001:**
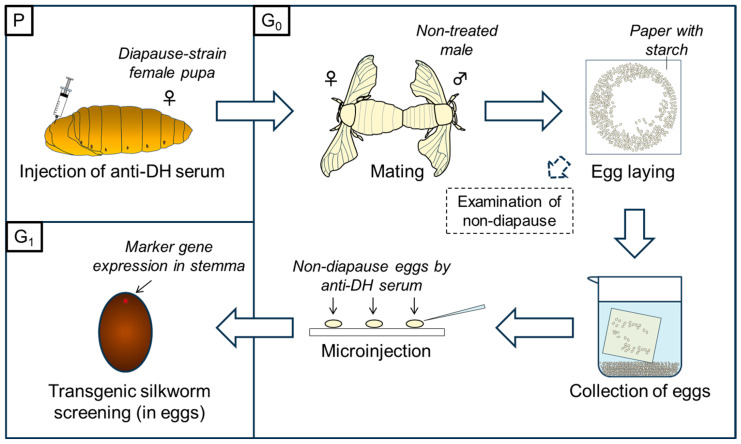
The outline of production of non-diapause eggs and transgenic silkworms. The outline is separated by three parts, P, G_0_, and G_1_ where “P” and “G” of them means “parental” and “generation,” respectively. In the depiction of P, the serum is injected at the thorax by the syringe; see the detail in Figure A1. All or parts of the eggs laid after aDHST were examined for the productivity of non-diapause. The eggs in G_0_ after aDHST become white because of the effect of the anti-BmDH serum, while the eggs in G_1_ become colored again without the effect of the anti-serum (see Figure 2).

**Figure 2 ijms-26-07604-f002:**
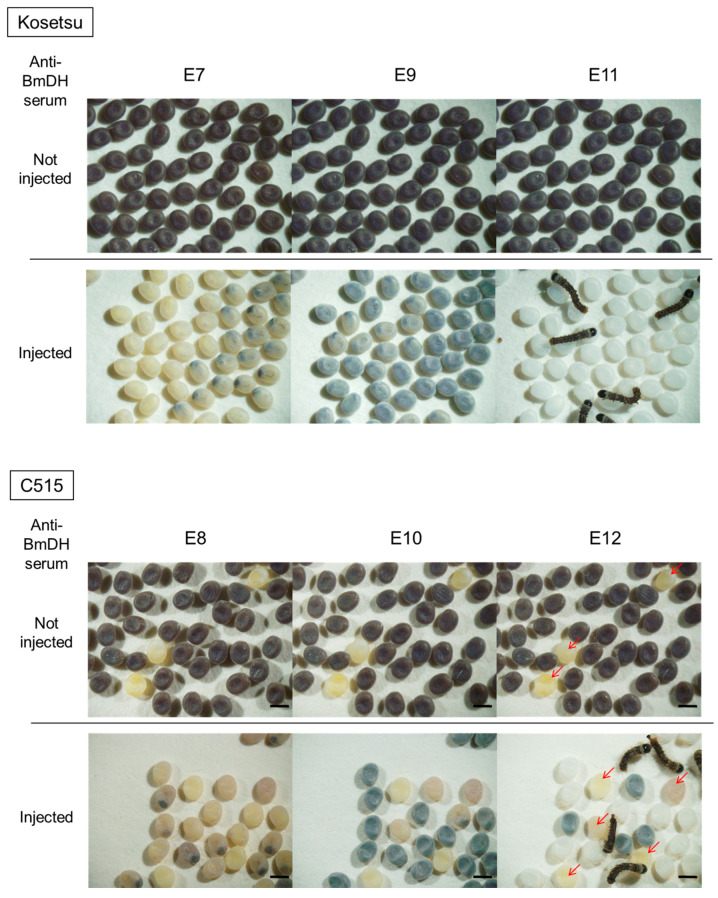
Non-diapause activity of eggs laid by female moths after aDHST. Kosetsu and C515 strains were used as host silkworm strains in this experiment. The photographs of eggs derived from pupae injected or not injected with anti-BmDH serum were taken in the same positions at each different developmental stage. E7, E8, E9, E10, E11, and E12 indicate the egg phases at 7, 8, 9, 10, 11, and 12 days after egg laying, respectively. The eggs of both strains turned white by aDHST. The blue color shown in E9 and E10 of the eggs injected with the serum was caused by melanin pigmentation in the larvae inside the eggshells. Red arrows indicate undeveloped eggs. Scale bar is 1 mm.

**Figure 3 ijms-26-07604-f003:**
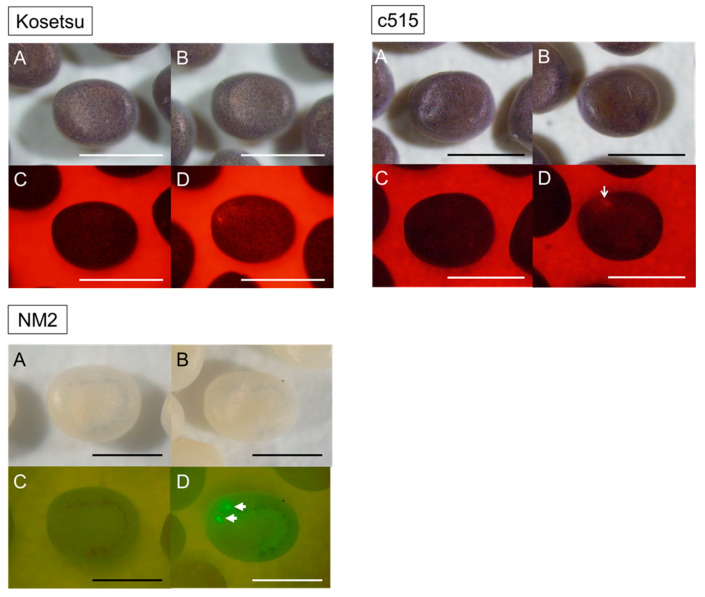
Transgenic silkworms in G_1_ generation. The Kosetsu, C515, and MN2 strains were used as host strains in this experiment. The photographs show the wild type (**A**,**C**) and the transgenic eggs (**B**,**D**) at 7 or 8 days after the eggs were laid. (**A**,**B**) show a bright view, and (**C**,**D**) show a fluorescent view. White arrows in (**D**) indicate the expression of the red or green fluorescent marker gene in the stemma. Scale bar is 1 mm.

**Figure 4 ijms-26-07604-f004:**
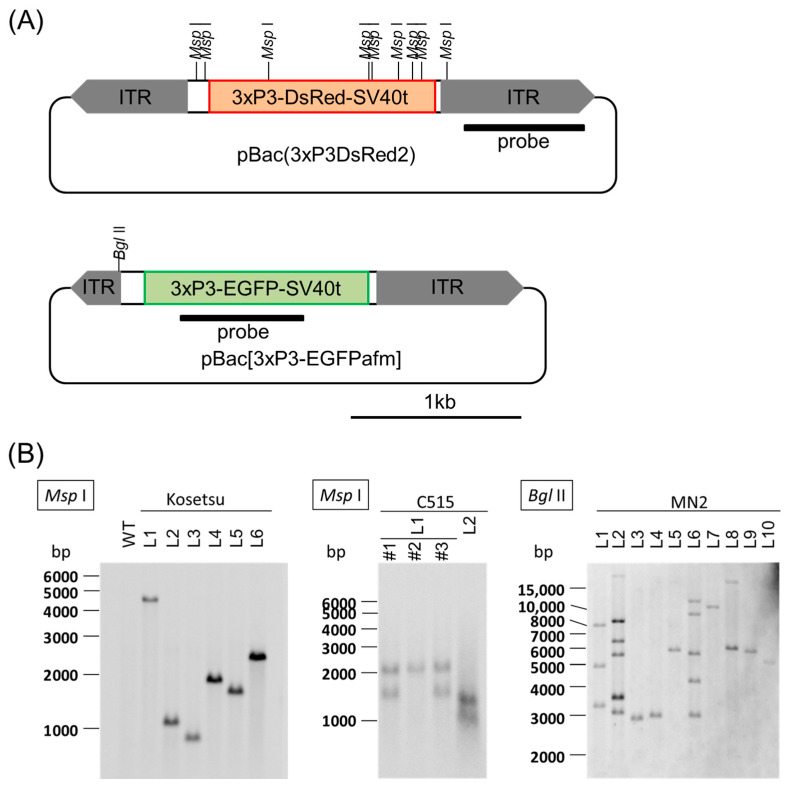
Southern blotting analysis. (**A**) The illustration depicts the vectors pBac(3xP3DsRed2), used in Kosetsu and C515, and pBac{3xP3-EGFPafm}, used in MN2, where the positions of *Msp* I cut sites, *Bgl* II cut site, and the probe used in the Southern blotting are described. L- and R-ITR indicate an inverted terminal repeat of the left arm and the right arm, respectively. (**B**) Southern blotting analysis in the Kosetsu, C515, and MN2 strains. Each lane (L) indicates individuals derived from different G_1_ broods expressing transgene markers: *3xP3DsRed2* and *3xP3-EGFP*. “#1–3” indicates different individuals in the same brood (L1) in C515. WT: wild type of non-transgenic individual, used as a negative control.

**Table 1 ijms-26-07604-t001:** Frequency of hatchability of progeny eggs after treatment at different temperatures in parental embryonic development.

Strains	Temperature in Parental Embryonic Development	No. of Broods Examined in G_0_ (n=)	(a) No. of Eggs Fertilized in G_0_ (Mean ± SE)	(b) No. of Eggs Hatched in G_0_ Broods (Mean ± SE)	No. of Diapause Eggs in G_0_ (Mean ± SE)	No. of Eggs Died in G_0_ (Mean ± SE)	Percentage of Eggs Hatched in Fertilized Eggs in G_0_ (b/a × 100)
Kosetsu	15 °C	5	1073 (215 ± 133)	743 (149 ± 88)	0 (0 ± 0)	330 (66 ± 52)	69.2%
	25 °C	4	2066 (517 ± 19)	2 (1 ± 1)	2050 (513 ± 20)	14 (4 ± 3)	0.1%
C515	15 °C	5	1376 (275 ± 142)	0 (0 ± 0)	1339 (268 ± 142)	37 (7 ± 4)	0%
	25 °C	5	1628 (326 ± 177)	0 (0 ± 0)	1569 (314 ± 169)	59 (12 ± 20)	0%

“G_0_” means a 1st progeny generation to parents.

**Table 2 ijms-26-07604-t002:** Hatchability of progeny eggs derived from female pupae injected with anti-BmDH serum.

Strains	Anti-BmDH Serum Injection	No. of Broods Examined in G_0_ (n=)	No. of Diapause Eggs in G_0_ Broods (Mean ± SE)	No. of Eggs Died in G_0_ Broods (Mean ± SE)	(a) No. of Eggs Hatched in G_0_ Broods (Mean ± SE)	(b) No. of Eggs Fertilized in G_0_ Broods (Mean ± SE)	Percentage of Eggs Hatched in Fertilized Eggs (a/b × 100)
Kosetsu	Not injected	9	4005 (445 ± 87)	73 (8 ± 10)	2 (0 ± 1)	4080 (453 ± 82)	0.05%
	Injected ^†^	13	7 (1 ± 0)	492 (38 ± 35)	4449 (342 ± 94)	4948 (381 ± 103)	89.9%
C515	Not injected	10	4017 (402 ± 121)	173 (17 ± 28)	0 (0 ± 0)	4190 (419 ± 119)	0%
	Injected ^†^	12	230 (19 ± 33)	711 (59 ± 26)	2248 (187 ± 53)	3189 (266 ± 60)	70.5%

^†^ These experiments were performed using eggs that were not used in the embryonic microinjection in Table 3; both experiments were performed using the eggs derived from the same female moths.

**Table 3 ijms-26-07604-t003:** Efficiencies of producing transgenic silkworms.

Host Strain	Vector (Final Consentration)	G_0_		G_1_	
		(a) No. of Eggs Injected	(b) No. of Eggs Hatched (b/a × 100)	(c) No. of Broods Examined	(d) No. of Broods with Transgene (d/c × 100)
Kosetsu	pBac(3xP3DsRed2) (0.2 μg/μL)	190	145 (76.3%)	59	6 (10.2%)
C515	pBac(3xP3DsRed2) (0.1 μg/μL), pBacIE1-Nat3xP3EGFP (0.1 μg/μL)	280	115 (41.1%)	35	2 (5.7%), 0 (0%) ^†^
MN2	pBac{3xP3-EGFPafm} (0.2 μg/μL)	194	121 (62.4%)	21	15 (71.4%)

^†^ The embryonic microinjection in the C515 strain was performed using two vectors, pBac(3xP3DsRed2) and pBacIE1-Nat3xP3EGFP, and then transgenic silkworms were produced only in pBac(3xP3DsRed2) but not in pBacIE1-Nat/3xP3EGFP.

## Data Availability

The datasets supporting the conclusions of this article are included within the article and its Appendix A.

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
