# Peer review of "Production of Transgenic Silkworm Using Anti-Serum Against Diapause Hormone in Diapause Strains of Silkworm, Bombyx mori"

_ijms, 2025, doi:10.3390/ijms26157604_

Round 1
Reviewer 1 Report
Comments and Suggestions for Authors
The study demonstrates significant innovation through its use of anti-diapause hormone serum (aDHS) to modify diapause characteristics in silkworm strains, offering considerable potential for advancing transgenic silkworm production and sericulture enhancement. However, several critical concerns regarding the interpretation of core data in Table 2 require author clarification:
(1) Definition of "Number of eggs fertilized":
We understand this to represent eggs demonstrating successful oocyte-spermatozoa fusion with subsequent developmental potential. Could the authors explain why the Kosetsu strain exhibits only 7 fertilized eggs among 13 egg-laying females?
(2) Definition of "Number of eggs hatched in G0 broods":
Should this be interpreted as the count of eggs (from both aDHST and control groups) that successfully bypassed diapause and completed direct hatching?
(3) Data consistency in hatch rate calculation (Column 8):
The methodology for deriving "Percentage of eggs hatched in fertilized eggs" from Columns 6 (diapauses eggs) and 7 (eggs died) requires clarification.
We respectfully request the authors to: Specify the operational criteria for "fertilized eggs" (e.g., cytological verification versus morphological evaluation);Provide the precise mathematical formula employed for hatch rate calculation
Additional Minor Queries Requiring Clarification:
1. Line 13: The current phrasing appears grammatically incomplete. Would "For conversion of egg diapause traits from diapause to non-diapause types" better convey the intended meaning?
2.Line 42: The "20h" parameter needs explicit definition. We recommend stating "20 hours post-oviposition" to eliminate temporal ambiguity.
3. Line 289: The reported aDHS injection volumes (7-90μl) demonstrate substantial variability. This appears inconsistent with the standardized protocol (10 or 90μl) described in Lines 127-129. Please: Explain the scientific basis for this volume range; Clarify the allocation criteria for different experimental groups.
4. Lines 291-293: What is the specific biological rationale for maintaining mated moths at 5°C for 1-3 days? Is this protocol: Primarily for oviposition synchronization, or an essential component of the post-pupal aDHS injection procedure? We would appreciate either a physiologically grounded explanation or supporting references from the literature.
Author Response
Comments 1:[Comments and Suggestions for Authors. The study demonstrates significant innovation through its use of anti-diapause hormone serum (aDHS) to modify diapause characteristics in silkworm strains, offering considerable potential for advancing transgenic silkworm production and sericulture enhancement. However, several critical concerns regarding the interpretation of core data in Table 2 require author clarification:]
Response 1:[Thank you very much for your polite review, especially pointing out on Table 2. Please review the revised manuscript again. Together with the author comments here, please see the revised file with making the revised places yellow (ijms-3705025_revision_1st with yellowed revisions_250702). Finally, the English of revised manuscript has been edited using MDPI Author Services. We are happy for you to reconsider our manuscript.]
Comments 2:[2 Definition of "Number of eggs fertilized": We understand this to represent eggs demonstrating successful oocyte-spermatozoa fusion with subsequent developmental potential. Could the authors explain why the Kosetsu strain exhibits only 7 fertilized eggs among 13 egg-laying females?]
Response 2:[Very sorry and thank you so much for your pointing out. As a result of confirming, there are mistakes in the heading and a part of numbers in Table 2. We corrected those. Please see yellow parts in Table 2.]
Comments 3:[Definition of "Number of eggs hatched in G0 broods": Should this be interpreted as the count of eggs (from both aDHST and control groups) that successfully bypassed diapause and completed direct hatching?]
Response 3:[As the above, we mistook the placement of each heading. Your interpretation is right. Please see the revised Table2.]
Comments 4:[Data consistency in hatch rate calculation (Column 8): The methodology for deriving "Percentage of eggs hatched in fertilized eggs" from Columns 6 (diapauses eggs) and 7 (eggs died) requires clarification. We respectfully request the authors to: Specify the operational criteria for "fertilized eggs" (e.g., cytological verification versus morphological evaluation);Provide the precise mathematical formula employed for hatch rate calculation]
Response 4:[Same as above. The calculation formula is “a/b*100” written in the column 8 of Table 2. Please see the revised Table2.]
Comments 5:[Line 13: The current phrasing appears grammatically incomplete. Would "For conversion of egg diapause traits from diapause to non-diapause types" better convey the intended meaning?]
Response 5:[Thank you for your polite suggestion. We completely accepted your proposal and revised. Please see the revised manuscript.]
Comments 6:[Line 42: The "20h" parameter needs explicit definition. We recommend stating "20 hours post-oviposition" to eliminate temporal ambiguity.]
Response 6:[Thank you for your pointing out. That is right. We revised. Please see the revised manuscript.]
Comments 7:[Line 289: The reported aDHS injection volumes (7-90μl) demonstrate substantial variability. This appears inconsistent with the standardized protocol (10 or 90μl) described in Lines 127-129. Please: Explain the scientific basis for this volume range; Clarify the allocation criteria for different experimental groups.]
Response 7:[Thank you very much. I think your pointing out is very important. The broad range of aDHS injection volumes in this study is mainly caused by aDHS titer: production lot. Although we used the two sera, among them Dr Shiomi’s one has a very high effectiveness. In a different issue, there is a difference of sensitivity to aDHS in diapause strains. Ultimately, there is no other option but to test it when aDHS are produced.]
Comments 8:[Lines 291-293: What is the specific biological rationale for maintaining mated moths at 5°C for 1-3 days? Is this protocol: Primarily for oviposition synchronization, or an essential component of the post-pupal aDHS injection procedure? We would appreciate either a physiologically grounded explanation or supporting references from the literature.]
Response 8:[Thank you for your question. It is empirically true that the maintaining mated (female) moth at low temperature easily lays as we have done (ref. 23, 37, 41). However, it may be not necessary of being 5 °C, which is not essential for the aDHST, but for embryonic microinjection.]
Reviewer 2 Report
Comments and Suggestions for Authors
Overall, the research findings provide a new method for the transgenic application of diapause strains in silkworms. The author demonstrates innovation and offers a novel solution for transgenic manipulation in diapause varieties. I have the following three questions.
- This method utilizes an anti-serum against the diapause hormone to counteract its induction effect on embryonic diapause in offspring. This step is crucial as it directly determines the proportion of diapausing embryos in the progeny. However, the paper merely mentions injecting 7 to 90 microliters of anti-serum per female pupa without specifying the optimal volume and concentration. Considering that different varieties produce varying levels of diapause hormone, the authors should specify the exact injection parameters (volume and concentration) of the anti-diapause hormone serum used for different varieties in this study.
- Theoretically speaking, the eggs laid by the univoltine and bivoltine strains after injection (G0 generation) would still produce diapausing G1 eggs. At this stage, should embryonic diapause be terminated by injecting anti-serum or through hydrochloric acid treatment? Additionally, would the accumulation of melanin in the serosa of diapausing eggs affect fluorescence screening?
- Currently, there are several existing methods for the transgenic application of diapause strains in silkworms, such as corona treatment, which appears more convenient. Have you compared these methods? What are the advantages of injecting anti-serum?
- I recommended that certain figures should retain ABCD labeling to facilitate reader comprehension. Many figures lack subpanel divisions with captions clustered together, detracting from clarity.
Author Response
Comments 1:[Comments and Suggestions for Authors. Overall, the research findings provide a new method for the transgenic application of diapause strains in silkworms. The author demonstrates innovation and offers a novel solution for transgenic manipulation in diapause varieties. I have the following three questions.]
Response 1:[Thank you for your polite review. Please review the revised manuscript again. Together with the author comments here, please see the revised file with making the revised places yellow (ijms-3705025_revision_1st with yellowed revisions_250702). Finally, the English of revised manuscript has been edited using MDPI Author Services. We are happy for you to reconsider our manuscript.]
Comments 2:[This method utilizes an anti-serum against the diapause hormone to counteract its induction effect on embryonic diapause in offspring. This step is crucial as it directly determines the proportion of diapausing embryos in the progeny. However, the paper merely mentions injecting 7 to 90 microliters of anti-serum per female pupa without specifying the optimal volume and concentration. Considering that different varieties produce varying levels of diapause hormone, the authors should specify the exact injection parameters (volume and concentration) of the anti-diapause hormone serum used for different varieties in this study.]
Response 2:[Thank you for your pointing out. We agree that ability for production and secretion of diapause hormone may differ in each strain. It’s also true that female pupae have different sizes in strains and even in individuals. Additionally, it should be considered that the titer of sera may differ in production lots. We used the serum including polyclonal antibody to diapause hormone peptide which can control the volume, but not the concentration; rather the concentration in the production lot of the sera should be considered the same. This paper aim is to show that anti-diapause hormone sera is useful for transgenesis via microinjection in diapause strains of Bombyx mori. Therefore, we demonstrated the effectiveness of making non-diapause eggs by anti-diapause hormone sera for production of transgenic silkworms in diapause strains, and that it works depending on the volume of the sera. However, the information of the serum volume in Table 2 was lacked, so we added in Line 83 of the manuscript that “we injected the serum of 7 μL per a female pupa of Kosetsu and C515 at the 1st day after ecdysis.”]
Comments 3:[Theoretically speaking, the eggs laid by the univoltine and bivoltine strains after injection (G0 generation) would still produce diapausing G1 eggs. At this stage, should embryonic diapause be terminated by injecting anti-serum or through hydrochloric acid treatment? Additionally, would the accumulation of melanin in the serosa of diapausing eggs affect fluorescence screening?]
Response 3:[Thank you for your question. We performed acid treatment to the progeny (G1) derived from microinjection to break diapause trait. We added about that in the section 4. 5 as “The collected eggs were immersed in an acid solution for 90 minutes: HCl diluted with distilled water, specific gravity of 1.1100 at 15 °C.” As you mentioned, the accumulation of melanin is a troublesome thing for the fluorescence screening as shown in Figure 3. The detection rate may be likelihood decrease by that.]
Comments 4:[Currently, there are several existing methods for the transgenic application of diapause strains in silkworms, such as corona treatment, which appears more convenient. Have you compared these methods? What are the advantages of injecting anti-serum?]
Response 4:[Thank you for your good question. We have not compared our method with the other methods. It's sure that the corona treatment may be as effective as our methods. However, the corona treatment requires a special equipment, and it is unclear whether the method is effective for insects other than silkworms. Although producing of sera costs but not needs special equipment. Additionally, we believe that our method may be effective in insects with embryonic diapause.]
Comments 5:[I recommended that certain figures should retain ABCD labeling to facilitate reader comprehension. Many figures lack subpanel divisions with captions clustered together, detracting from clarity.]
Response 5:[Thank you very much for your kind suggestion. You mean about Figure 2 and 3, right. Those didn’t deliberatively adopt ABCD labeling because we separately explain those results in each strain. Therefore, we attached the strain name to each result, especially, it is also a reason that ABCD labeling has already been used in each photograph of Figure 3.]
Reviewer 3 Report
Comments and Suggestions for Authors
This is an excellent study that uses an ingenious idea, blocking the action of the diapause hormone in vivo, to extend the usefulness of germline transformation to a wider collection of strains of the domesticated silkmoth.
I have several suggestions for minor corrections, and one or two questions, see below.
The issue of the patent must be cleared up. If the patent referred to concerns the research reported in the present manuscript, it must be declared in the conflict of interest section. This might apply even if the patent concerns previous research, which would accrue a financial benefit to the inventors by the publication of the manuscript. Please ensure that the policies of the journal are followed.
line 38, corpus cardiacum, not ccorpus ardacum
line 41, treating what? treating the eggs after oviposition?
line 45, in constant darkness, NOT no constant darkness
line 54, injecting, not injection
line 55, diapause, not diapaus
line 59, "does not require any special equipment". This is not correct; special equipment is required for the injections of the eggs for transformation. This equipment must be described.
line 80, list Dr. Shiomi's university or institute here.
line 96, what aspect of brown pigment production is induced by diapause hormone? Is it biosynthesis of the ommochromes, or transport of ommochrome precursors into the pigment granules?
List of abbreviations for aDHST: diapause, not diapause
Figure 1 shows the injection location. The injection location on the pupa must be explicitly stated in the text and Figure A1 must be explicitly mentioned in the text. Currently there is no reference to Figure A1 in the text.
line 84, laid, not lied
Figure 2. the middle panel shows a blue color instead of dark brown for injected eggs. Is the blue also due to ommmochromes? Or is there some other blue pigment that is masked by the dark brown ommochromes in diapause eggs?
Table 2 heading, diapause eggs, not diapauses eggs
line 141. Why isn't the larval phenotype, light brown instead of black, shown as a result of injection of pBac[IE1Nat3xP3GFP]? Only the egg phenotypes are shown. Also, there are no Southern blots for that plasmid. Did that part of the transformation fail for some reason? Please comment.
line 267. from, not form
line 281. commercial, not trustee
line 286. what animal was injected to produce the antibodies? Rabbit? mouse? horse?
line 327. More detail about the patent is needed. What invention does it cover, one described in a previous publication or the present one? Filing a patent based on the findings in the present publication raises a financial conflict of interest, and this must be reported in the Conflicts of Interest section,
Comments on the Quality of English Language
The manuscript needs to be proofread by a native English speaker or a professional proofreading service. There are many small errors, of which I have corrected only a fraction.
Author Response
Comments 1:[Comments and Suggestions for Authors. This is an excellent study that uses an ingenious idea, blocking the action of the diapause hormone in vivo, to extend the usefulness of germline transformation to a wider collection of strains of the domesticated silkmoth. I have several suggestions for minor corrections, and one or two questions, see below.]
Response 1:[Thank you for your polite review. Please review the revised manuscript again. Together with the author comments here, please see the revised file with making the revised places yellow (ijms-3705025_revision_1st with yellowed revisions_250702). Finally, the English of revised manuscript has been edited using MDPI Author Services. We are happy for you to reconsider our manuscript.]
Comments 2:[The issue of the patent must be cleared up. If the patent referred to concerns the research reported in the present manuscript, it must be declared in the conflict of interest section. This might apply even if the patent concerns previous research, which would accrue a financial benefit to the inventors by the publication of the manuscript. Please ensure that the policies of the journal are followed.]
Response 2:[Thank you for your pointing out on the patent. As mentioned in the section of “6. Patents,” we have already obtained the patent on this study and declared that there is not any conflict of interests in the manuscript because the patent holder all are included in this paper as co-authors. We added that into the section of “6. Patents,” Please see.]
Comments 3:[line 38, corpus cardiacum, not ccorpus ardacum]
Response 3:[Thank you very much. I corrected it. Please see in the manuscript.]
Comments 4:[line 41, treating what? treating the eggs after oviposition?]
Response 4:[Thank you for your pointing out. We revised as following; “Embryonic diapause induction can be suppressed by immersion of HCl solution at around 20 hours after eggs laying.” Please see in the manuscript.]
Comments 5:[line 45, in constant darkness, NOT no constant darkness]
Response 5:[Thank you very much. We corrected it. Please see in the manuscript.]
Comments 6:[line 54, injecting, not injection]
Response 6:[Thank you very much. We corrected it. Please see in the manuscript.]
Comments 7:[ine 55, diapause, not diapaus]
Response 7:[Thank you very much. We corrected it. Please see in the manuscript.]
Comments 8:[line 59, "does not require any special equipment". This is not correct; special equipment is required for the injections of the eggs for transformation. This equipment must be described.]
Response 8:[Thank you for your pointing out. We revised as following; “Our method using ant-diapause hormone sera is highly efficient, does not require any special equipment excluding conventional microinjection devices, and makes it a practical approach.” Please see in the manuscript.]
Comments 9:[line 80, list Dr. Shiomi's university or institute here.]
Response 9:[Thank you for your suggestion. We add his affiliation, “Shinshu university.” Please see in the manuscript.]
Comments 10:[line 96, what aspect of brown pigment production is induced by diapause hormone? Is it biosynthesis of the ommochromes, or transport of ommochrome precursors into the pigment granules?]
Response 10:[Thank you for your good question. Although I imagine that the biosynthesis of ommochromes is responsible for it, I don’t know why on the detail about that. However, it is empirically known that the eggs color turns blown to white when they don’t become diapause in Bombyx mori.]
Comments 11:[List of abbreviations for aDHST: diapause, not diapause]
Response 11:[Sorry, I can’t understand this meaning.]
Comments 12:[Figure 1 shows the injection location. The injection location on the pupa must be explicitly stated in the text and Figure A1 must be explicitly mentioned in the text. Currently there is no reference to Figure A1 in the text.]
Response 12:[Thank you very much for your pointing out. We added the explanation of the injection location and Figure A1 in the legend of Figure 1 as following: “In the depict of G-1, the serum is injected at the thorax by the syringe, see the detail in Figure A1.”]
Comments 13:[line 84, laid, not lied]
Response 13:[Thank you very much. We corrected it. Please see in the manuscript.]
Comments 14:[Figure 2. the middle panel shows a blue color instead of dark brown for injected eggs. Is the blue also due to ommmochromes? Or is there some other blue pigment that is masked by the dark brown ommochromes in diapause eggs?]
Response 14:[Thank you for your good question. The blue color is caused of melanin pigmentation in the larvae inside of the eggshells, not ommochromes. The ommochromes are accumulated in the serosa membrane of eggs which are finally taken in by larva. We added the explanation on the blue color in the Figure 2 legend. Please see in the manuscript.]
Comments 15:[Table 2 heading, diapause eggs, not diapauses eggs]
Response 15:[Thank you very much. We corrected it and in Table 1 as well. Please see in the manuscript.]
Comments 16:[line 141. Why isn't the larval phenotype, light brown instead of black, shown as a result of injection of pBac[IE1Nat3xP3GFP]? Only the egg phenotypes are shown. Also, there are no Southern blots for that plasmid. Did that part of the transformation fail for some reason? Please comment.]
Response 16:[Thank you for your good question. The color change is caused of the expression of IE1Nat gene. Please see the detail in the reference 39. We performed co-microinjection of two vector, pBac(3xP3DsRed2) and pBac[IE1Nat3xP3GFP], but unfortunately, we couldn’t obtained any transgenic silkworms in the pBac[IE1Nat3xP3GFP] as mentioned in the Table 3 legend.]
Comments 17:[line 267. from, not form]
Response 17:[Thank you very much. We corrected it. Please see in the manuscript.]
Comments 18:[line 281. commercial, not trustee]
Response 18:[Thank you very much. We corrected it. Please see in the manuscript.]
Comments 19:[line 286. what animal was injected to produce the antibodies? Rabbit? mouse? horse?]
Response 19:[Thank you very much for your pointing out. That is rabbits. We added that in the applicable part. Please see in the manuscript.]
Comments 20:[line 327. More detail about the patent is needed. What invention does it cover, one described in a previous publication or the present one? Filing a patent based on the findings in the present publication raises a financial conflict of interest, and this must be reported in the Conflicts of Interest section,]
Response 20:[Thank you for your pointing out. We have already responded about this issue in your comment #1. Please see.]
Round 2
Reviewer 1 Report
Comments and Suggestions for Authors
I fully understand and trust that the author has provided reasonable responses to the mentioned issues. The study's approach of using anti-diapause hormone serum to alter the diapause characteristics of silkworm strains holds significant value for the improvement of sericulture production varieties. From the perspective of pupal injection, excessive injection volumes may also cause harm to individual specimens. Standardizing the quality control of the diapause hormone serum would further enhance its practical application.
Author Response
Comments 1:[I fully understand and trust that the author has provided reasonable responses to the mentioned issues. The study's approach of using anti-diapause hormone serum to alter the diapause characteristics of silkworm strains holds significant value for the improvement of sericulture production varieties. From the perspective of pupal injection, excessive injection volumes may also cause harm to individual specimens. Standardizing the quality control of the diapause hormone serum would further enhance its practical application.]
Response 1:[Thank you for your kind comments and helpful suggestions. We will continue to research careful consideration to the standardization of quality control of the diapause hormone serum.]
Reviewer 2 Report
Comments and Suggestions for Authors
The author has addressed my concerns, and I believe the manuscript can be published in its current form.
Author Response
Comments 1:[The author has addressed my concerns, and I believe the manuscript can be published in its current form.]
Response 1:[Thank you for your kind comments and helpful suggestions.]